# Site-Specific Management Zones Delineation Based on Apparent Soil Electrical Conductivity in Two Contrasting Fields of Southern Brazil

Eduardo Leonel Bottega [1,*], José Lucas Safanelli [2,3], Mojtaba Zeraatpisheh [4,5], Telmo Jorge Carneiro Amado [6,†], Daniel Marçal de Queiroz [7] and Zanandra Boff de Oliveira [1]

1   Federal University of Santa Maria (UFSM), Cachoeira do Sul Campus, Cachoeira do Sul 96506-322, RS, Brazil; zanandra.oliveira@ufsm.br
2   College of Agriculture "Luiz de Queiroz", University of São Paulo, Piracicaba 13416-000, SP, Brazil; jose.lucas.safanelli@usp.br or jsafanelli@ woodwellclimate.org
3   Woodwell Climate Research Center, Falmouth, MA 02540, USA
4   Rubenstein School of Environment and Natural Resources, University of Vermont, 81 Carrigan Drive, Burlington, VT 05405, USA; mojtaba.zeraatpisheh@uvm.edu
5   Gund Institute for Environment, University of Vermont, 210 Colchester Ave, Burlington, VT 05401, USA
6   Center for Rural Sciences (CCR), Soil Department, Federal University of Santa Maria (UFSM), Santa Maria 97105-900, RS, Brazil; amado@ksu.edu
7   Agricultural Engineering Department (DEA), Federal University of Viçosa (UFV), Viçosa 36570-900, MG, Brazil; queiroz@ufv.br
*   Correspondence: eduardo.bottega@ufsm.br
†   Current address: Agronomy Department, Kansas State University, Manhattan, KS 66506, USA.

**Abstract:** Management practices that aim to increase the profitability of agricultural production with minimal environmental impact must consider within-field soil variability, and this site-specific management can be addressed by precision agriculture (PA). Thus, this work aimed to investigate which key soil attributes are distinguishable management zones (MZ) delineated based on the soil apparent electrical conductivity (ECa), using fuzzy k-means, in two fields with contrasting soil textures in southern Brazil. For this, a grid scheme (50 × 50 m) was applied to measure ECa, conduct soil sampling for analysis, and determine soybean yield. The MZ were delineated based on the ECa spatial distribution, and statistical non-parametric tests ($p < 0.05$) were employed to compare the soil chemical and physical attributes among MZ. The management zones were able to distinguish the average values of Clay, Silt, pH, $Ca^{2+}$, $Mg^{2+}$, SB, $Al^{3+}$, $H^+ + Al^{3+}$, AS%, and BS%. In the field classified as sandy clay loam texture, management zones were able to differentiate the average values of soybean yield, Clay, $Ca^{2+}$, $Mg^{2+}$, SB, and CEC. Thus, this study supports the ECa as an efficient tool for delineating MZ of contrasting cropland soils in southern Brazil to understand the within-field soil variability and adjust the inputs according.

**Keywords:** precision agriculture; soil nutrients; clay soil; sandy clay loam



## 1. Introduction

Brazil has a wide range of climates, topography, vegetation, and soils as a country of continental dimensions. Combined with the use of technologies, these characteristics make it possible for agriculture intensification that, in some regions with favorable rainfall distribution and temperature, allow the production of up to three crops of grains in the same field year-round [1].

This fact helps explain the agricultural year of 2019/2020 when Brazil broke a record in grain production, totaling 257.8 million tons. The soybean is highlighted in this context, as it presented the production record, estimated at 124.8 million tons with a 4.3% increase compared to the 2018/19 season [1]. This placed Brazil as the world's largest producer of soybean.

The southern region of Brazil is responsible for producing 79.5 million tons of grain, representing 30.8% of the national production in the 2019/2020 agricultural year. The states of Santa Catarina and the Rio Grande do Sul contributed 53.7% of the production obtained in the region [1].

The improvements in crop production are as a result of the increased use of technologies, such as the no-till system, the genetic modification of cultivars, the evolution of agricultural machines, and, more recently, data-driven management based on the collection, analysis, and mapping of georeferenced information, especially those associated with the attributes from soil and crop performance. Adopting these agricultural practices has allowed more assertive management, contributing to the better use of inputs, reducing costs, and increasing productivity, making Brazilian agriculture increasingly competitive.

Understanding the spatial variability of soil properties is of critical importance for precision agriculture (PA) and food security, water security, and to combat land degradation [2]. The ability to manage the soil resource relies on the scale at which one can observe and model the soil's characteristics and its processes. In this sense, PA is a tool that aims to increase the profitability of agricultural production with minimal environmental impact based on the management of within-field spatial variability, either from soils, crops, or the local climate. However, the direct and traditional methods employed in PA demand a large density of samples in order to accurately describe the within-field soil variability [3].

Grid schemes are both resource-intensive and time-consuming, being not economically feasible for small fields or due to logistical and time issues in large fields, as both the temporal and spatial scales affect the soil attributes' variability [4]. Site-specific management zones, or simply management zones (MZ), are field subdivisions that share similar characteristics that drive crop yield or support the homogeneous application of crop inputs, such as seeds, fertilizers, and agrochemicals. This is a broad approach for managing the spatial variability of crops and soils [5–7], one of PA's and digital agriculture's fundamental principles [8].

The success of soil MZ depends on the relationship between the soil attributes and the indirect measurements used for spatial delineation. Measuring the soil's apparent electrical conductivity (ECa) is fast and inexpensive compared to traditional soil sampling and laboratory analysis [3,9]. Several studies have demonstrated a strong correlation between ECa and soil physicochemical attributes, such as soil texture, moisture, and some nutrients that affect crop yields [10–12]. However, the performance of ECa for MZ delineation is affected not only by within-field soil variability, but also by the contrasting characteristics of production fields. This fact can make the processes governing the relationship between ECa and the soil variability determined by each agro-eco-region.

Studies are still scarce on the use of ECa data as a tool in decision-making on the management to be used in tropical soils. In these soils, the use of ECa mapping can reduce costs with soil sampling to recommend correctives and fertilizers, enabling localized applications and in precise doses. Research has already shown that detecting the spatial variability of soil attributes and determining management zones using soil apparent electrical conductivity (ECa) sensors have helped to reduce the costs associated with soil sampling [13,14].

Another possible application is the use of ECa mapping as a tool in the management of the soybean plant population. Studies conducted in the Brazilian Cerrado region observed that the apparent electrical conductivity and clay content variables showed greater correlation and direct effects on the soybean yield [15]. The authors concluded that areas with higher electrical conductivity values and clay content should receive higher populations of soybean plants. Thus, this work aimed to investigate which key soil attributes are distinguishable between the MZ of two contrasting fields in southern Brazil, contributing to the understanding of soils and their impact on soybean yield.

## 2. Materials and Methods

### 2.1. Experimental Areas

The present study was carried out in two commercial fields of grain production, which have different soil textural characteristics. The first (Field C) is located in the Curitibanos municipality, state of Santa Catarina. In contrast, the second (Field SCL) is located in the Cachoeira do Sul municipality, state of Rio Grande do Sul. Field C has 13 hectares and has its central geographical coordinates located at 27°20′33″ S and 50°34′23″ W. The climate is classified as humid mesothermal with mild summer (Cfb), with an average annual temperature of 16 °C and an average rainfall of 1400 to 1600 mm per year [16]. The average altitude of the area with sea level is 1026 m.

Field SCL covers 25.8 hectares and has its geographical coordinates located at 30°17′24″ S and 53°00′44″ W. The climate is classified as humid subtropical with hot summer (Cfa), with an annual average temperature of 20 °C and average rainfall ranging from 1600 to 1900 mm per year [16]. The average altitude of the area to sea level is 134 m. According to the USDA [17], the average textural classification of the areas Field C and Field SCL are clay and sandy clay loam, respectively (Figure 1).

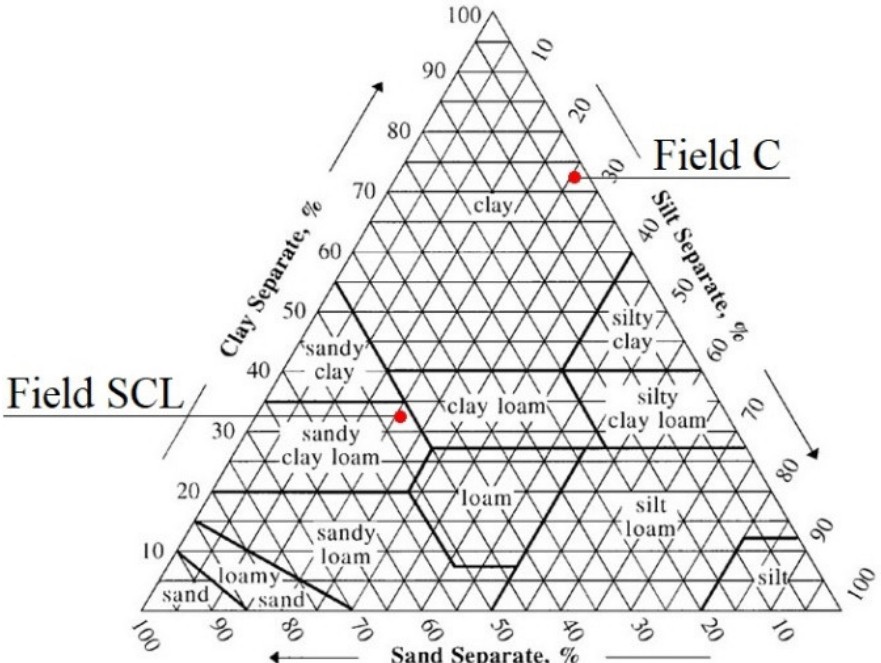

**Figure 1.** Textural soil classification of the two study areas [17]. Source: https://www.nrcs.usda.gov/wps/portal/nrcs/detail/soils/survey/?cid=nrcs142p2_054167 (accessed on 8 May 2022).

### 2.2. Apparent Soil Electrical Conductivity, Soil Samples, and Soybean Grain Yield

Sampling grids were used as a positioning reference for measuring the ECa, collecting topsoil samples (0–20 cm) for laboratory analysis, and estimating the soybean yield. In Field C, a sampling grid composed of 50 points (sampling density of approximately 3.8 points per hectare, Figure 2A) was established. In contrast, in Field SCL, the sampling grid consisted of 95 points (sampling density of approximately 3.7 points per hectare). The sample points were spaced at 50 × 50 m in both of the sites. A Garmin GPS receiver, model GPSMAP 62sc, was used for navigation in the areas. In Field C, soil samples were taken at all 50 sample points. In Field SCL soil samples were taken at only 24 out of 95 sample points, as represented in Figure 2B.

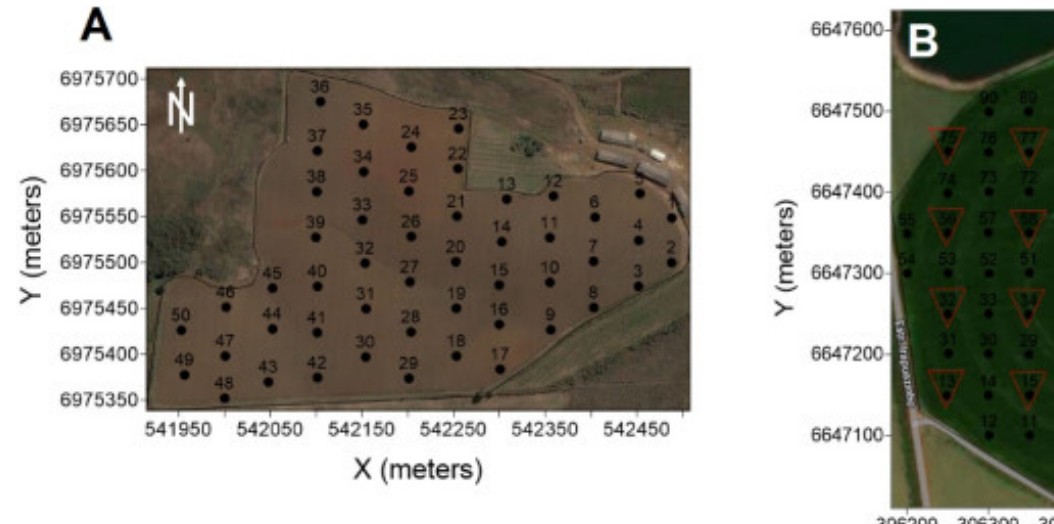
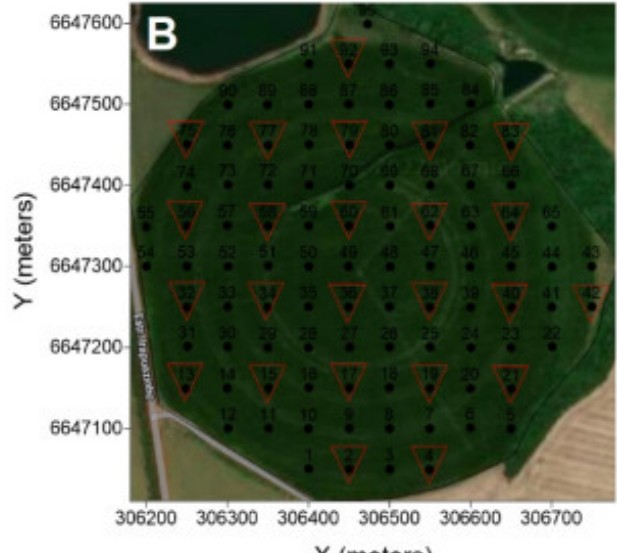

**Figure 2.** Representative map of experimental areas and sampling points. Field classified with clay texture (**A**) located in Curitibanos/SC (Field C) and field classified with sandy clay loam texture (**B**) located in Cachoeira do Sul/RS (Field SCL). Triangle in (**B**) indicates soil sampling.

The soil sampling was carried out by taking five subsamples within a radius of 5 m around the coordinate point, representing the topsoil at a depth of 0–20 cm. The subsamples were mixed and homogenized to make a composite sample, which was defined as representative of the point. In this task, a Dutch-type auger was used for manually sampling the soils. The composite samples were sent to the laboratory for standard chemical and physical soil analysis, which included the Clay, Sand, and Silt content, soil organic matter, pH (water), bases ($Ca^{2+}$, $Mg^{2+}$, $K^+$), P, $Al^{3+}$, and $H^+ + Al^{3+}$ [18]. The soil moisture was determined by the standard gravimetric method: drying at 110 °C for 24 h, using part of the soil sampled at each point.

At each sampling point, ECa was measured using the portable LandMapper ERM-02 conductivity meter (Landviser®, League City, TX, USA). This device applies an electric current to the external electrodes and measures the potential difference in the internal electrodes. The electrodes were organized according to the Wenner Matrix [19,20]. The support structure of the electrodes was developed using metalon tubes, steel screws, and flexible wires in different colors for the two current electrodes (red wire) and two potential electrodes (black wire). The contact between the steel screws and the metalon was isolated, covering the screws with a PVC hose. Figure 3 shows the ERM-02 set and the Wenner Matrix.

The resistivity obtained using the Wenner Matrix was calculated by Equation (1):

$$\rho = \frac{2 \cdot \pi \cdot \alpha \cdot \Delta V}{i} \tag{1}$$

where $\rho$ is the resistivity (Ohm m$^{-1}$); $\alpha$ is the electrode spacing (m); $\Delta V$ is the potential difference (V), and $i$ is the applied electrical current (A).

The ECa represents the inverse of resistivity, is calculated using Equation (2):

$$ECa = \frac{1}{\rho} \tag{2}$$

where ECa is the apparent electrical conductivity (mS m$^{-1}$). The georeferenced values of ECa and soil analysis were prepared for further analysis.

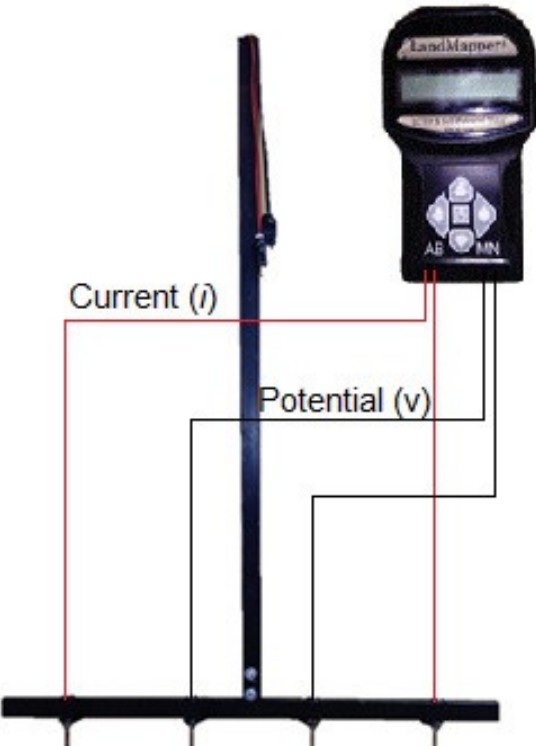

**Figure 3.** Wenner Matrix and portable device, LandMapper® ERM-02.

### 2.3. Geostatistical Analysis

The spatial dependence of ECa was assessed by adjusting variograms, assuming the stationarity of the intrinsic hypothesis, defined by Equation (3):

$$\hat{\gamma}(h) = \frac{1}{2N(h)} \sum_{i=1}^{N(h)} [Z(x_i) - Z(x_i + h)]^2 \tag{3}$$

where $\hat{\gamma}(h)$ is the semi-variance as a function of the separation distance $(h)$ between pairs of points; $h$ is the separation distance between pairs of points, $m$; and $N(h)$ is the number of experimental pairs of $Z(x_i)$ and $Z(x_i + h)$ observations separated by a distance $h$.

The following variogram models were tested: linear with sill; Gaussian; Spherical; and Exponential models. The model that resulted in the smallest residual sum of squares (RSS) by 10-fold cross-validation was selected to represent the theoretical variogram. The following parameters were determined in the analysis: nugget effect ($C_0$), sill ($C_0 + C$), and range (A).

The interpolation was performed in a 5-m grid using the ordinary Kriging method. This interpolation method was selected because it provides the best linear bias forecasts (BLUP), seeking the minimum variance [21]. For the estimates of values in non-sampled locations, 16 close neighbors and a search radius equal to 50% of the range value found in the variogram adjustment were used in order to guarantee the spatial continuity interval. The software GS+ version 7 [22] was used for the geostatistical analysis, while the Surfer software, version 10 [23], was employed for making the thematic maps.

### 2.4. Management Zones and Statistical Analysis

The management zones were generated from the interpolated ECa maps. In this task, the KrigMe software was employed. The software uses the Fuzzy k-means algorithm as a clustering method for generating the MZ. The ECa data were grouped into two distinct classes, resulting in field maps with two MZ (MZ1 and MZ2) [24].

The sampling grids with soil results were superimposed on the interpolated maps of MZ to compose the dataset used for statistical analysis. Sampling points remarkably close to the limits of each MZ were not considered for the statistical analysis, thereby reducing the errors associated with spatial transitions. A descriptive analysis of the mean, minimum, maximum, and standard deviation values was initially calculated. Subsequently, the attributes were compared using the non-parametric Mann–Whitney test ($p < 0.05$). Statistical analyzes were performed using the R software version 4.04 [25].

## 3. Results

The field with a clay texture (Field C) had average soil moisture of 0.23 $m^3$ $m^{-3}$ when measuring ECa, while the field classified as sandy clay loam had average soil moisture of 0.14 $m^3$ $m^{-3}$. The ECa had the spatial variability confirmed by variogram analysis in the two studied areas (Table 1). The theoretical model adjusted to the empirical semi-variance of the ECa was spherical, a common model employed in similar studies [13,26]. This model represents low spatial continuity; that is, small changes in the ECa values are observed from one point to another. The highest range (467.8 m) was observed for the area with sandy clay loam texture. This parameter determines the search radius used in interpolating non-sampled locations; thus, estimates that use larger range values tend to be more reliable [27].

**Table 1.** Theoretical model parameters were adjusted to the empirical semi-variance of the ECa in the two study areas.

| Field | Geostatistical Parameters | | | | | |
|---|---|---|---|---|---|---|
| | **Model** | **A** [1] | **$C_0 + C$** [2] | **$C_0$** [3] | **RSS** [4] | **$R^2$** [5] |
| **C** | Spherical | 111.00 | 47.19 | 2.80 | 223 | 0.47 |
| **SCL** | Spherical | 467.80 | 5.93 | 2.72 | 0.105 | 0.98 |
| | Cross-validation parameters | | | | | |
| | **Regression coefficient** | | **Y** [6] | | **SEE** [7] | **$R^2$** |
| **C** | 0.88 | | 2.58 | | 5.81 | 0.22 |
| **SCL** | 0.91 | | 0.97 | | 1.95 | 0.22 |

[1] Range (m); [2] Sill; [3] Nugget effect; [4] Residual sum of the square; [5] Coefficient of determination; [6] Intercept; [7] Standard error of the estimate.

Figure 4 shows the spatial variability of ECa in both of the experimental areas (Figure 4A,B) and the resulting maps of management zones with the superimposed soil sampling points (Figure 4C,D). Circled points in red were not used for the comparison test, as their location was close to the transition boundaries of the management zones. This procedure was completed to reduce the uncertainty associated with the interpolation of ECa. For the study area Field C, MZ1 was characterized by 24 points, while 21 sample points were considered for MZ2. For the study area Field SCL, the characterization of MZ1 was constituted by five points, whereas MZ2 considered 12 sampling points. Although a few points were used for MZ1 in Field SCL, a smaller deviation was observed for 9 out of the 15 soil attributes evaluated (Table 2).

After the descriptive analysis, the minimum values of ECa for MZ1 were close in both of the studied areas. A great difference was obtained for the maximum values regardless of the MZ, with 45.45 mS $m^{-1}$ and 13.42 mS $m^{-1}$ for Field C and Field SCL, respectively (Tables 2 and 3). This difference can be explained by two factors, i.e., the higher clay content and the higher soil moisture of Field C when measuring ECa. Field C presented the highest average values for chemical attributes of the soil, especially for macronutrients, such as $Ca^{2+}$, $Mg^{2+}$, and $K^+$.

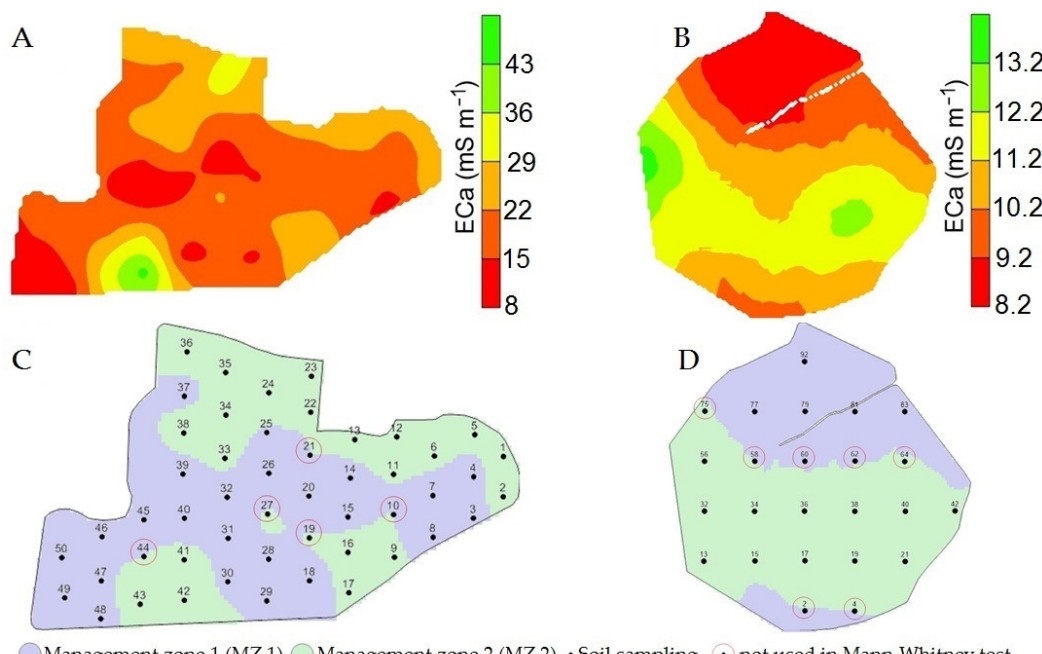

**Figure 4.** Spatial variability of ECa at the fields with clay (**A**) and sandy clay loam textures (**B**). Management zone maps (MZ1 and MZ2) and soil sampling points for clayey (**C**) and sandy clay loam fields (**D**).

**Table 2.** Descriptive statistics of soil attributes for the sandy clay loam area ("Field SCL").

| Soil Attributes | Management Zone 1 (MZ 1) | | | | | Management Zone 2 (MZ 2) | | | | |
|---|---|---|---|---|---|---|---|---|---|---|
| | Valid N | Mean | Minimum | Maximum | Std. Dev. | Valid N | Mean | Minimum | Maximum | Std. Dev. |
| ECa | 5 | 9.52 | 8.28 | 10.23 | 1.84 | 12 | 11.47 | 10.85 | 13.42 | 3.05 |
| Soybean | 5 | 4563.25 | 4451.97 | 4636.95 | 70.90 | 12 | 4308.82 | 3755.41 | 5474.38 | 461.30 |
| Clay | 5 | 30.60 | 26.00 | 37.00 | 4.39 | 12 | 35.50 | 33.00 | 37.00 | 1.57 |
| Sand | 5 | 48.60 | 41.00 | 55.00 | 5.59 | 12 | 45.08 | 41.00 | 50.00 | 2.91 |
| Silt | 5 | 20.80 | 19.00 | 22.00 | 1.30 | 12 | 19.42 | 15.00 | 22.00 | 2.23 |
| pH | 5 | 5.08 | 4.70 | 5.40 | 0.26 | 12 | 4.98 | 4.40 | 5.60 | 0.43 |
| SOM | 5 | 2.14 | 1.70 | 2.50 | 0.30 | 12 | 2.38 | 2.10 | 2.70 | 0.17 |
| P | 5 | 53.26 | 22.00 | 72.90 | 18.92 | 12 | 37.25 | 14.20 | 82.40 | 20.73 |
| $K^+$ | 5 | 162.20 | 110.00 | 204.00 | 36.99 | 12 | 195.25 | 140.00 | 310.00 | 47.64 |
| $Al^{3+}$ | 5 | 0.22 | 0.10 | 0.50 | 0.16 | 12 | 0.26 | 0.00 | 0.70 | 0.22 |
| $H^+ + Al^{3+}$ | 5 | 5.32 | 4.10 | 6.40 | 0.93 | 12 | 5.10 | 3.00 | 8.60 | 1.87 |
| $Ca^{2+}$ | 5 | 3.92 | 3.50 | 5.10 | 0.66 | 12 | 5.65 | 4.60 | 6.90 | 0.68 |
| $Mg^{2+}$ | 5 | 1.66 | 1.33 | 2.16 | 0.31 | 12 | 2.02 | 1.62 | 2.54 | 0.29 |
| SB | 5 | 6.00 | 5.10 | 7.80 | 1.05 | 12 | 8.18 | 6.90 | 9.90 | 0.96 |
| CEC | 5 | 6.24 | 5.60 | 7.90 | 0.95 | 12 | 8.44 | 7.40 | 10.00 | 0.83 |
| BS | 5 | 53.12 | 44.40 | 61.30 | 6.94 | 12 | 62.35 | 46.20 | 76.80 | 10.22 |
| AS | 5 | 3.84 | 1.30 | 8.80 | 2.98 | 12 | 3.25 | 0.00 | 8.80 | 2.75 |

SB: Sum of bases; AS: Aluminum saturation; BS: Base saturation; ECa: mS m$^{-1}$; Soybean: kg ha$^{-1}$; Clay, Sand, Silt, SOM, BS, and AS: %; P and $K^+$: mg L$^{-1}$; $Al^{3+}$, $H^+ + Al^{3+}$, $Ca^{2+}$, $Mg^{2+}$, SB, and CEC: cmol$_c$ L$^{-1}$.

The highest average values of the attributes of soil organic matter (SOM), potassium ($K^+$), calcium ($Ca^{2+}$), magnesium ($Mg^{2+}$), the sum of bases (SB), cation exchange capacity (CEC), and base saturation (BS) were observed in MZ2, for both of the areas studied. MZ2 was characterized as the MZ with the highest mean value of ECa. The highest mean values of the attributes' potential acidity ($H^+ + Al^{3+}$) and aluminum saturation (AS) were observed for MZ1, which presented the lowest mean values of ECa for both areas. These results may indicate that sites with higher ECa have better soil fertility. However, this is just a hypothesis.

**Table 3.** Descriptive statistics of soybean yield and soil attributes for the clay area (Field C).

| Soil Attributes | Management Zone 1 (MZ 1) | | | | | Management Zone 2 (MZ 2) | | | | |
|---|---|---|---|---|---|---|---|---|---|---|
| | Valid N | Mean | Minimum | Maximum | Std. Dev. | Valid N | Mean | Minimum | Maximum | Std. Dev. |
| ECa | 24 | 15.23 | 8.00 | 18.93 | 2.78 | 21 | 26.44 | 21.27 | 45.45 | 5.38 |
| Soybean | 24 | 4221.91 | 3257.33 | 5295.20 | 577.01 | 21 | 4061.01 | 3141.33 | 5525.20 | 599.20 |
| Clay | 24 | 74.28 | 65.50 | 85.50 | 4.46 | 21 | 69.81 | 63.00 | 78.00 | 3.78 |
| Sand | 24 | 2.99 | 2.50 | 3.70 | 0.30 | 21 | 3.00 | 2.00 | 5.00 | 0.63 |
| Silt | 24 | 22.73 | 12.00 | 31.50 | 4.34 | 21 | 27.57 | 19.00 | 35.00 | 3.94 |
| pH | 24 | 5.58 | 4.90 | 6.30 | 0.32 | 21 | 6.09 | 5.40 | 6.90 | 0.38 |
| SOM | 24 | 3.51 | 3.10 | 4.30 | 0.32 | 21 | 3.60 | 2.80 | 4.30 | 0.39 |
| P | 24 | 54.99 | 20.00 | 95.00 | 20.18 | 21 | 53.71 | 22.00 | 122.00 | 28.59 |
| $K^+$ | 24 | 120.71 | 62.40 | 198.90 | 41.16 | 21 | 134.46 | 58.50 | 261.30 | 58.50 |
| $Al^{3+}$ | 24 | 0.09 | 0.00 | 1.03 | 0.21 | 21 | 0.00 | 0.00 | 0.09 | 0.02 |
| $H^+ + Al^{3+}$ | 24 | 6.14 | 3.70 | 9.40 | 1.39 | 21 | 4.18 | 2.30 | 6.90 | 1.17 |
| $Ca^{2+}$ | 24 | 7.49 | 4.60 | 9.40 | 1.22 | 21 | 9.11 | 7.40 | 11.20 | 1.07 |
| $Mg^{2+}$ | 24 | 2.30 | 1.50 | 3.30 | 0.43 | 21 | 3.13 | 2.40 | 4.30 | 0.57 |
| SB | 24 | 10.09 | 6.30 | 12.50 | 1.58 | 21 | 12.60 | 10.20 | 15.60 | 1.60 |
| CEC | 24 | 16.22 | 14.30 | 18.70 | 1.07 | 21 | 16.78 | 14.80 | 19.00 | 1.37 |
| BS | 24 | 62.11 | 40.00 | 77.00 | 8.61 | 21 | 75.00 | 61.00 | 85.00 | 6.71 |
| AS | 24 | 1.15 | 0.00 | 14.00 | 2.91 | 21 | 0.05 | 0.00 | 1.00 | 0.22 |

SB: Sum of bases; AS: Aluminum saturation; BS: Base saturation; ECa: mS m$^{-1}$; Soybean: kg ha$^{-1}$; Clay, Sand, Silt, SOM, BS, and AS: %; P and $K^+$: mg L$^{-1}$; $Al^{3+}$, $H^+ + Al^{3+}$, $Ca^{2+}$, $Mg^{2+}$, SB and CEC: cmol$_c$ L$^{-1}$.

Figure 5 shows the result of the Mann–Whitney test. For Field C, of the 17 attributes studied, the delimitation of MZ was able to differentiate the average values of 11. They include ECa, Clay, Silt, pH, $Ca^{2+}$, $Mg^{2+}$, SB, $Al^{3+}$, $H^+ + Al^{3+}$, AS, and BS. As for Field SCL, of the 17 attributes studied, seven showed differences between the averages depending on the management zones to which they belong, including Soybean yield, ECa, Clay, $Ca^{2+}$, $Mg^{2+}$, SB, and CEC. These results demonstrate that the delimitation of MZ, based on the spatial variability of ECa, tends to differentiate a greater number of attributes in soils with a high clay content. It was observed that, regardless of the clay content of the studied area, the soil attributes of ECa, Clay, $Ca^{2+}$, $Mg^{2+}$, and SB presented significantly different mean values, depending on the sampling by management zones.

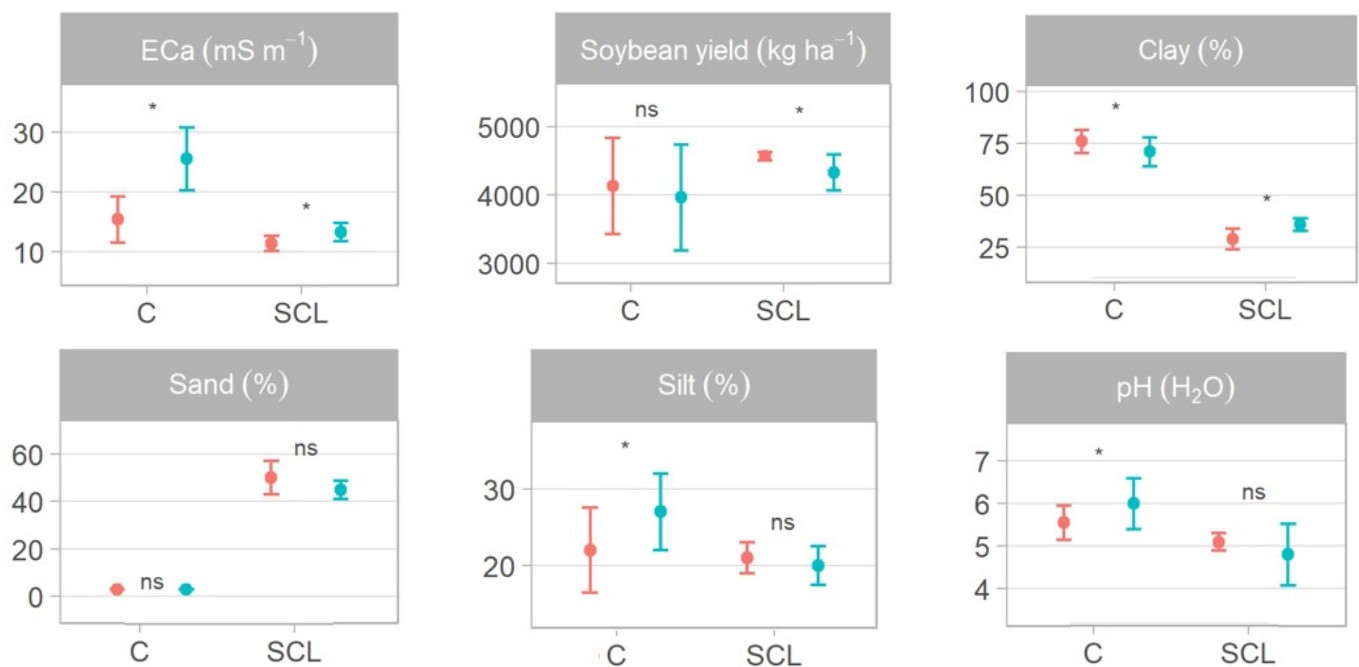

**Figure 5.** *Cont.*

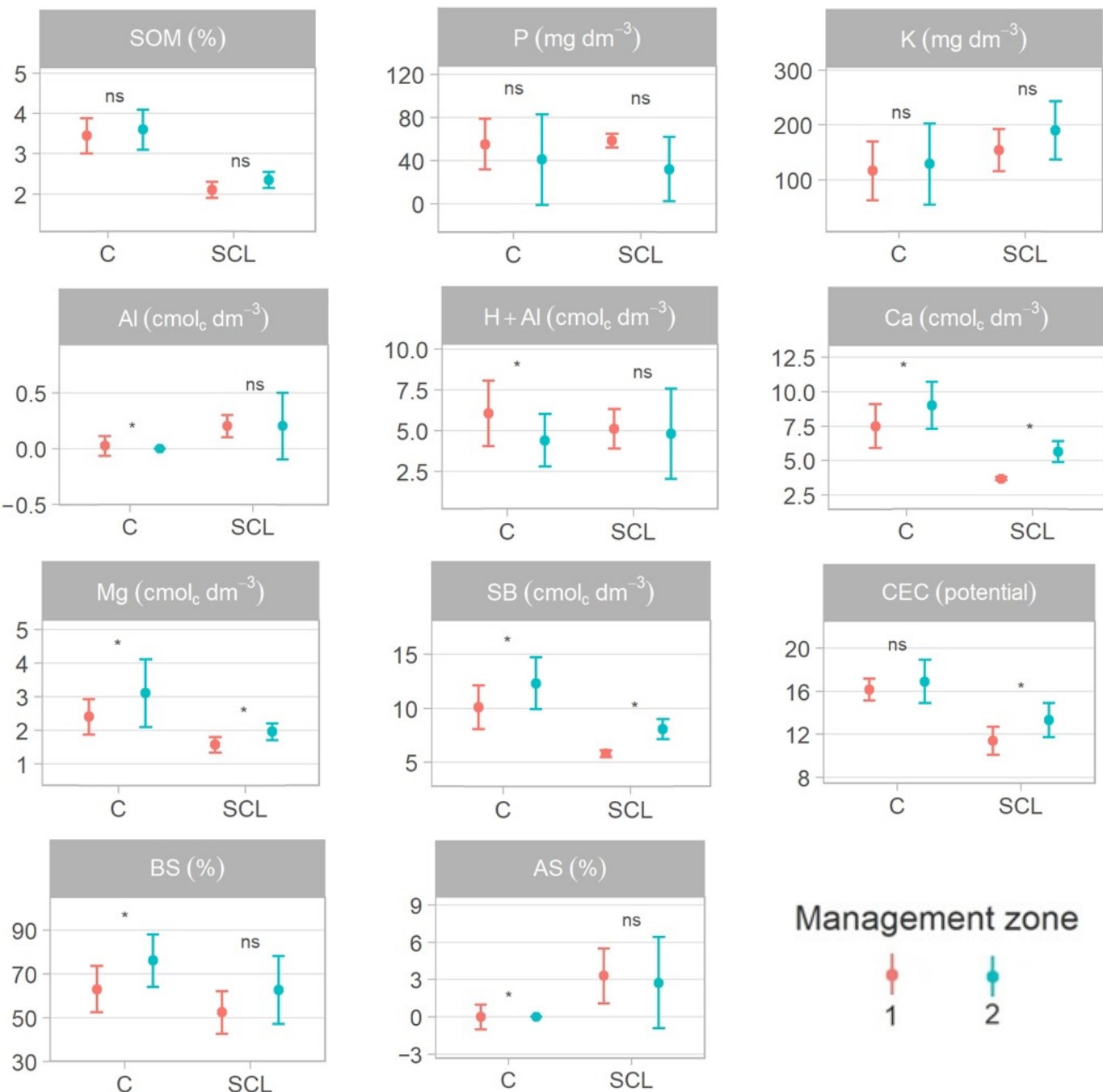

**Figure 5.** Mann–Whitney test results ($p < 0.05$) for attribute averages as a function of management zones. Note: C: clay texture; SCL: sandy clay loam texture. (*) Mann–Whitney test results significant. (ns) Not significant.

## 4. Discussion

Due to their higher microporosity, a higher moisture content is expected for clay soil in relation to sandy clay loam soil. According to Michelon et al. [28], there is a high relationship between microporosity and soil water retention. In this respect, Brevik et al. [29] highlight that soil ECa is controlled by multiple factors, such as clay mineralogy, clay content, and soil moisture.

The highest standard deviation values were observed for soybean grain yield, P, and $K^+$ (Tables 2 and 3). These results can be explained by the direct relationship between the availability of these nutrients in the soil and soybean yield. Edaphoclimatic conditions

associated with chemical soil management, especially phosphate and potassium fertilizers, influenced the yield components of the soybean crop, such as the number of pods per plant, the number of seeds per pod, and the specific seed weight [30].

In the soybean, the deficiency of phosphorus (P) reduces the yield potential due to the lower production of flowers and pods, higher rate of abortion of these structures, and the production of seeds with a lower weight [31]. Potassium ($K^+$) deficiency can affect the opening and closing of stomata and water efficiency use, causing the production of lighter seeds, resulting in lower and less productive plants [32]. Serafim et al. [33] observed an increase in grain yield and the specific weight of seeds in response to fertilization with this nutrient [34].

In both of the fields, MZs differentiated average values of Clay, $Ca^{2+}$, $Mg^{2+}$, and SB, demonstrating that, even in agricultural areas with contrasting textural classification, these attributes greatly influence ECa. This fact can be explained by the relationship of these attributes with how soil conducts electricity. The soil presents three forms of electrical current conduction: (1) liquid phase, through the amount of water and dissolved solids present in the soil macropores; (2) solid–liquid phase, dependent on exchangeable cations associated with clay minerals; and (3) solid phase, based on the proximity between soil particles [20].

The behavior of nutrient availability due to pH variations, mainly $Ca^{2+}$ and $Mg^{2+}$ (represented by BS), varies depending on the clay content of the soil [35]. In the present study, the MZ delimited for both fields were able to differentiate the contents of $Ca^{2+}$, $Mg^{2+}$, and SB. However, in Field SCL, MZ could not differentiate the pH and BS. The ability to exchange cations, together with pH, has a great relationship with the saturation value by soil bases and plant nutrient availability [36]. Therefore, any other soil attribute that causes changes in these relationships will also affect the soil–plant relationship.

By differentiating the average values of BS% among the MZs, based on the mapping of the ECa, soils with a high clay content could be used to manage soil acidity, allowing the recommendation of lime application, based on the chemical characterization of each MZ. According to Costa et al. [37], base saturation (BS%) represents an attribute of great agronomic value, as it is used as a calculation basis to recommend corrective measures to neutralize acidity and exchangeable Al in the soil. The authors found a high correlation between this soil attribute and ECa; for this reason, they highlighted that the use of ECa maps in the delimitation of soil MZ is noteworthy.

The highest CEC values were observed for the soil with the lowest clay content (Field SCL), whereas the highest SOM contents were observed for the field with the highest clay content (Field C). In the study conducted by Soria et al. [35], the authors demonstrated that an increase in the clay content resulted in a higher CEC value for the same SOM value when comparing sandy and clayey soils. In tropical soils, the soil organic matter is responsible for 75 to 90% of the CEC of the soil [38]. Therefore, the increase in this attribute increases the amount of cations that the soil will be able to retain [39]. Sana et al. [36] highlight that the cation exchange capacity, together with pH, has a great relationship with the saturation value by soil bases and its availability of nutrients to plants.

In a study conducted in Argentina by Peralta et al. [40], comparing two agricultural fields with different textures (sandy and sandy-loam), the authors observed that the delimited MZ, based on the measurement of ECa, were able to differentiate the contents of SOM, $Mg^{2+}$, $Ca^{2+}$, pH, and CEC, as in the present study. The authors concluded that ECa measurements successfully delimited two homogeneous soil zones associated with the spatial distribution of soil properties, which could support the establishment of more efficient sampling schemes.

Considering the attributes differentiated by the Mann–Whitney test (Figure 5), it was observed that in Field C (high clay content), the highest mean value of ECa (Table 3) was observed for the MZ2 that presented the lowest value of attributes: $Al^{3+}$, $H^+ + Al^{3+}$, and AS% (Table 3). This same MZ presented a higher mean value of the attributes: pH, $Ca^{2+}$, $Mg^{2+}$, SB, and BS% (Table 3). Similar behavior was observed for Field SCL (less clay),

where the MZ2 with the highest average value of ECa (Table 2) was also the one that presented the highest average values of the attributes $Ca^{2+}$, $Mg^{2+}$, SB, and CEC (Table 2), apparently indicating that sites with higher ECa values may be associated with sites that present greater soil fertility.

A fact that draws attention is that the MZ with the highest average value of ECa, in Field C, presented the lowest average value of clay, whereas, in Field SCL, the highest average value of ECa was obtained in the MZ with the highest clay content. A hypothesis that can be raised is that in clayey soils, the influence of the electrical charge of the clay minerals on the current conduction through the soil is suppressed by the electrical charge of the soil solution that had salts dissolved in it. In soils with lower clay content, the influence of clay minerals on the conduction of electric current is greater and fast drainage could move down the salts. However, for this behavior to be proven, a more detailed study would be necessary.

## 5. Conclusions

Spatial variability of the soil's apparent electrical conductivity was detected at two subtropical fields. In the experimental area classified as clay texture (Field C), the management zones defined, based on the spatial variability of ECa, were able to differentiate the average values of Clay, Silt, pH, $Ca^{2+}$, $Mg^{2+}$, SB, $Al^{3+}$, $H^+ + Al^{3+}$, AS%, and BS%. In the experimental area classified as sandy clay loam texture (Field SCL), management zones could differentiate the average values of soybean yield, Clay, $Ca^{2+}$, $Mg^{2+}$, SB, and CEC. Thus, our study supports the proposition of using ECa to delineate the MZ of contrasting agricultural soils in southern Brazil, reducing the need for high-density samplings to understand the within-field soil variability. The recommendation and application of fertilizers, based on soil sampling in management zones, should be the objective of future studies, to quantify how this form of management influences the yield performance of agricultural crops. These future studies are needed to validate this way of managing soil fertility.

**Author Contributions:** Conceptualization, E.L.B., J.L.S. and D.M.d.Q.; methodology, E.L.B., J.L.S., T.J.C.A. and D.M.d.Q.; software, E.L.B. and J.L.S.; validation, E.L.B., J.L.S. and Z.B.d.O.; formal analysis, E.L.B., J.L.S. and Z.B.d.O.; funding acquisition, E.L.B.; investigation, E.L.B., J.L.S., D.M.d.Q. and Z.B.d.O.; project administration, E.L.B.; supervision, E.L.B.; writing—original draft, E.L.B., J.L.S. and M.Z.; writing—review and editing, D.M.d.Q., T.J.C.A., M.Z. and Z.B.d.O. All authors have read and agreed to the published version of the manuscript.

**Funding:** This research received external funding from the National Council for Scientific and Technological Development (Notice n. 009/2019 Institutional Program for Scientific Initiation Scholarships–PIBIC/CNPq/UFSM) and the Agrisus Foundation (PA nº 1246/13) for granting scholarships.

**Institutional Review Board Statement:** Not applicable.

**Informed Consent Statement:** Not applicable.

**Data Availability Statement:** The data presented in this study are available on request from the corresponding author. The data are not publicly available for ethical reasons.

**Acknowledgments:** The authors thank the ConnectFarm Agricultural Consultancy LTDA for the logistical contributions in the development of this study. They also thank Kaoru Haramoto and Wilson Barufaldi for making the agricultural areas available.

**Conflicts of Interest:** The authors declare no conflict of interest.

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
