# Peer review of "Site-Specific Management Zones Delineation Based on Apparent Soil Electrical Conductivity in Two Contrasting Fields of Southern Brazil"

_agronomy, doi:10.3390/agronomy12061390_

Round 1
Reviewer 1 Report
Dear colleagues
I studied your paper
First of all, I must say that I find your work useful
The methods described are clearly a trend in global agriculture and have not only a local but a global impact.
I would like to make a few small remarks.
it would be good to add a map / picture of the experimental site to the methodology for a better idea.
on the contrary, the conductivity equations may be slightly redundant, but I do not insist on an adjustment.
within the results, especially the tables and graphs in Figure 5, it would be appropriate to highlight the statistically significant differences.
I consider the discussion and conclusion to be adequate.
I wish you much success in your future work.
Author Response
Dear Reviewer 1;
We appreciate your valuable contribution in making our study even more interesting.
All your suggestions for improvement were incorporated into the manuscript.
Attached we send a file with point-to-point responses.
Thanks.

Reviewer 2 Report
1. In terms of correct style of chapter headings/ references/ tables/ figure /units /decimals which are occasionally wrong formatted the authors can consult: journal recommendations for authors.
2. There are some type of typos in your MS. Please double check you text and correct them all.
3. The list of references should be standardized. Authors can read published papers by the journal in order to understand and follow the style.
4. Please add the specific problems and challenges of the findings (or what should be done in future) in the conclusion.
Author Response

(The authors gave the same response as above.)
